# Machine learning to predict mesenchymal stem cell efficacy for cartilage repair

**Yu Yang Fredrik Liu**[1]\*, **Yin Lu**[2], **Steve Oh**[2], **Gareth J. Conduit**[1]

**1** Theory of Condensed Matter Group, Cavendish Laboratory, University of Cambridge, Cambridge, United Kingdom, **2** Bioprocessing Technology Institute, Agency for Science Technology and Research (A*STAR), Singapore, Singapore

\* ly297@cam.ac.uk

**Data Availability Statement:** All data files are available from the Open Access database (https://doi.org/10.17863/CAM.52036).

**Funding:** Our Financial Disclosure statement has been amended to read: Y.Y.F.L. acknowledges

## Abstract

Inconsistent therapeutic efficacy of mesenchymal stem cells (MSCs) in regenerative medicine has been documented in many clinical trials. Precise prediction on the therapeutic outcome of a MSC therapy based on the patient's conditions would provide valuable references for clinicians to decide the treatment strategies. In this article, we performed a meta-analysis on MSC therapies for cartilage repair using machine learning. A small database was generated from published *in vivo* and clinical studies. The unique features of our neural network model in handling missing data and calculating prediction uncertainty enabled precise prediction of post-treatment cartilage repair scores with coefficient of determination of 0.637 ± 0.005. From this model, we identified defect area percentage, defect depth percentage, implantation cell number, body weight, tissue source, and the type of cartilage damage as critical properties that significant impact cartilage repair. A dosage of 17 − 25 million MSCs was found to achieve optimal cartilage repair. Further, critical thresholds at 6% and 64% of cartilage damage in area, and 22% and 56% in depth were predicted to significantly compromise on the efficacy of MSC therapy. This study, for the first time, demonstrated machine learning of patient-specific cartilage repair post MSC therapy. This approach can be applied to identify and investigate more critical properties involved in MSC-induced cartilage repair, and adapted for other clinical indications.

## Author summary

Cartilage damage affects the life quality of hundreds of millions of people, causing chronic joint pain and disability. Cartilage has poor regenerative capacity. Only minor damage could improve on its own or with passive treatments, while more severe damage often requires surgery. In recent decades, stem cell therapy has become a promising treatment option to reduce pain and repair cartilage. However, with complex mechanisms and various factors involved, efficient and consistent cartilage regeneration remains elusive. Our neural network learns information from clinical trials and animal studies to predict therapeutic outcomes along with the confidence level based on the patient's condition. This machine learning approach provides an important reference and significant insights into the optimization of treatment strategies.

support from the Agency for Science, Technology and Research for NSS scholarship. Y.L. and S.O. acknowledge the Agency for Science, Technology and Research for funding of this research through Bioprocessing Technology Institute. G.J.C. acknowledges support from the Royal Society with grant number UF130122. The funders had no role in study design, data collection and analysis, decision to publish, or preparation of the manuscript.

**Competing interests:** Y.Y.F.L. declares a potential financial conflict of interest as chief technology officer of DeepVerse. G.J.C. declares a potential financial conflict of interest as chief technology officer of Intellegens, UK.

## Introduction

Articular cartilage is a critical tissue with multifaceted mechanical functions. It holds compression, absorbs shock, and enables smooth articulation at the joints. Cartilage injury is unfortunately common due to tears, accidents and arthritis, which often leads to joint pain, stiffness, and inflammation. Cartilage disorders affect millions of people worldwide, including 52.2 million adults in US [1], and more than 10 million in UK [2]. In particular, osteoarthritis alone affects more than 200 million people globally [3]. Adult cartilage has limited self-repair capacity due to its avascular nature [4], thus treatments are often necessary to accelerate repair and relieve pain during joint motions. Besides the conservative treatments and conventional surgical options, such as microfracture and autologous chondrocyte implantation (ACI), mesenchymal stem cell (MSC) has also been widely investigated in the management of cartilage damages in recent decades [5].

Although significant success has been achieved for MSC therapy in cartilage repair, the efficacy of therapy has been inconsistent. This is likely attributed to the complex cellular mechanisms and dynamic interplay across different populations of cells involved in the stem cell assisted tissue repair processes. MSC therapy is also complicated by heterogeneity of cell, culture conditions, delivery methods, and recipients' conditions, which are all highly variable in current clinical trials and laboratory studies. Thus, disconnectedness between the *in vitro*, preclinical, and clinical performances of MSCs have been broadly observed [5], which has so far rendered the analysis of MSCs' therapeutic efficacy largely retrospective, rather than predictive. As a result, there is a lack of guidelines on MSC therapy strategy to promote optimal therapeutic efficacy.

Setting guidelines for MSC therapy requires identification of critical properties that affect MSCs' therapeutic efficacy most significantly. To achieve this, quantitative assessment of the significance of individual property is needed. However, this is ineffective through conventional controlled biomedical experiments where one or at most a few properties can be interrogated at a time. To overcome this challenge, we use machine learning to capture multi-property correlations and exploit all of the information in a database. A machine learning model predicts based on the training dataset, and each algorithm has a basic set of parameters to fit multidimensional functions that can be changed to improve its accuracy [6]. Deep learning methods are able to predict multiple output properties simultaneously [7].

In this paper, we performed a meta-analysis on MSC therapy for cartilage repair. The data we analyzed were generated by different researchers using different experimental designs; as a result, the properties considered in one study may not always be addressed in another, which has led to a database containing "missing information" in some of its entries. Many machine learning methods do not analyze the entries with incomplete information, which often results in a shrinking database with compromised cognitive performance. We adapt a neural network formalism [8–12] with a unique capacity to "fill" the missing data by learning the correlations across multiple properties, and recursively imputes with precise estimates. Furthermore, our machine learning method computes the uncertainty of predictions raised from experimental noise and computational extrapolation, which allows the neural network model to focus on the most confident predictions.

The coefficient of determination ($R^2$) of our machine learning model in predicting MSC therapy outcome was $0.637 \pm 0.005$ in cross-validation test. Through machine learning, we identified defect area percentage, defect depth percentage, implantation cell number, body weight, tissue source, and cartilage damage type as critical therapy properties of cartilage repair. In particular, an optimal dosage range of 17-25 million cells was identified for achieving the best therapeutic outcome. We also predicted that the optimal therapy outcome was most

likely to be achieved in patients with cartilage defects less than 6% in area and 22% in depth of the knee cartilage. Larger defects significantly dampen the efficacy of MSC therapy.

The capacity of predicting MSCs' therapeutic outcome using machine learning holds great clinical significance in suggesting critical therapy input properties to maximize the therapeutic benefits. Further development of this technology could extend its applications in other diseases and cell types, and shed light on substantial improvements in cell therapy efficacy and consistency.

## Methods

### Data sets

We collected data from 36 published articles on PubMed [13–48] to train and validate our machine learning models. Some articles comprised more than one type of cartilage injury models or treatment conditions. In total, 15 clinical trial conditions and 29 animal model conditions (1 goat, 6 pigs, 2 dogs, 9 rabbits, 9 rats, and 2 mice) on osteochondral injury or osteoarthritis were included, where MSCs were transplanted to repair the cartilage tissue. We documented each case into an entry of a database. We considered the cell- and treatment target-related factors as input properties, including species, body weight, tissue source, cell number, cell concentration, defect area, defect depth, and type of cartilage damage. The therapeutic outcomes were considered as output properties, which were evaluated using integrated clinical and histological cartilage repair scores, including the international cartilage repair society (ICRS) scoring system, the O'Driscoll score, the Pineda score, the Mankin score, the osteoarthritis research society international (OARSI) scoring system, the international knee documentation committee (IKDC) score, the visual analog score (VAS) for pain, the knee injury and osteoarthritis outcome score (KOOS), the Western Ontario and McMaster Universities Osteoarthritis Index (WOMAC), and Lyscholm score. In this study, these scores were linearly normalized to a number between 0 and 1, with 0 representing the worst damage or pain, and 1 representing the completely healthy tissue. The list of entries was combined together to form a database.

### Neural network formalism

We now define the neural network formalism that was used to capture the functional relation between all properties, and predict these relations for new therapies. The establishment of the core neural network and its critical feature on estimating the uncertainty in predictions are described as follow, before the second novel aspect of handling missing data.

Each entry $\mathbf{x} = (x_1, \ldots, x_I)$ to the neural network is a vector of length I, with the first $I - 1$ variables being the distinct treatment conditions (including species, body weight, tissue source, cell number, cell concentration, defect area, defect depth, and type of cartilage damage); and the final Ith variable is the therapeutic outcome. We intended to find a function $f$ that satisfies the fixed-point equation $\mathbf{f}(\mathbf{x}) \equiv \mathbf{x}$ for all entries in the database. The trivial solution to this fixed-point equation is the identity operator, $\mathbf{f}(\mathbf{x}) = \mathbf{x}$, but this solution does not allow us to impute data using the function $f$. We search for a solution to the fixed-point equation that is orthogonal to the identity operator, and allow the function to predict a given component of $\mathbf{x}$ from some or all other components. The output $(y_1, \ldots, y_I)$ is a vector of length I, with the first $I - 1$ variables being the predicted treatment conditions (if unknown); and the final Ith variable is the therapeutic outcome. A linear superposition of hyperbolic tangents was chosen to model the function $\mathbf{x}$,

$$\mathbf{f} : (x_1, \ldots, x_i, \ldots, x_I) \mapsto (y_1, \ldots, y_j, \ldots, y_I) \tag{1}$$

with

$$y_j = \sum_{h=1}^{H} C_{hj}\eta_{hj} + D_j$$

and

$$\eta_{hj} = \tanh\left(\sum_{i=1}^{I} A_{ihj}x_i + B_{hj}\right).$$

The neural network with one layer of hidden nodes was shown in Fig 1. Each hidden node $\eta_{hj}$ performed a tanh operation on a superposition of input properties $x_i$ with variational parameters $A_{ihj}$ and $B_{hj}$ for $1 \leq i \leq I$. Each property $y_j$ for $1 \leq j \leq I$ was predicted separately as a superposition of all hidden nodes with variational parameters $C_{hj}$ and $D_j$. There are exactly as many given properties as predicted properties, since all types of properties are treated equally

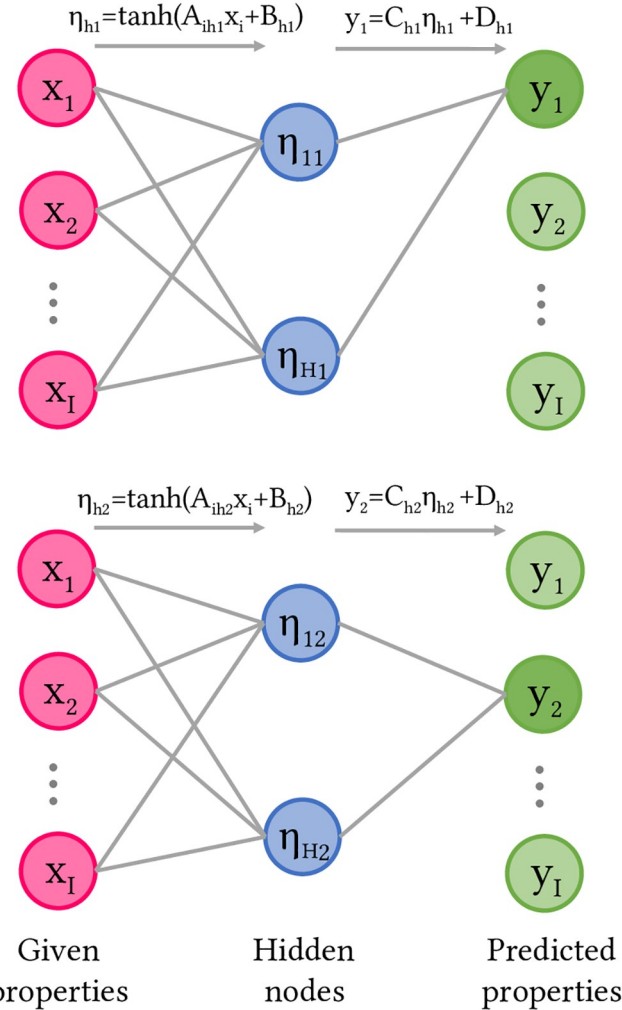

**Fig 1. Neural network to model data.** The graph illustrates how different inputs $x_i$ are used to calculate the outputs for $y_1$ (top) and $y_2$ (bottom); similar graphs can be drawn for all other $y_j$ to compute all the predicted properties. Linear combinations (grey lines) of the given properties (red) are taken by the hidden nodes (blue) through a non-linear tanh operation is applied, and a linear combination (grey lines) of the hidden nodes returns the predicted property (green).

by the ANN. We set the $A_{\text{jhj}} = 0$ so that the network predicts $y_j$ without knowledge of the true quantity $x_j$. A hyperbolic tangent activation function was used to constrain the magnitude of $\eta_{\text{hj}}$, giving the weights $C_{\text{hj}}$ sole responsibility for the amplitude of the output response. The variational parameters were selected to minimize the mean square error of predictions of the training data.

The ANN has to be trained on a provided data set. The parameters $\{A_{\text{ihj}}, B_{\text{hj}}, C_{\text{hj}}, D_j\}$ are initialized with random values, and varied following a random walk. The new values are accepted, if the new function $\mathbf{f}(\mathbf{x})$ models the fixed-point equation $\mathbf{f}(\mathbf{x}) \equiv \mathbf{x}$ better, which is quantitatively measured by the error function,

$$\sigma = \sqrt{\frac{1}{N} \sum_{x \in X} \sum_{j=1}^{I} [f_j(\mathbf{x}) - x_j]^2}. \tag{2}$$

This form is also known as the root-mean-square error (RMSE) cost function. The optimization is equivalent to the minimization of the RMSE cost function and a steepest descent approach is used.

In order to measure the uncertainty in the ANN's prediction, we train a number of models simultaneously, and treat their average as the overall prediction and their standard deviation as the uncertainty. The pseudocode is shown in Algorithm A, at least 100 training models were used to evaluate the uncertainty. This concept is similar when estimating the uncertainty in ensemble models, with the underlying model being changed to neural networks and the uncertainty generated accounts for both experimental uncertainty in the underlying data and the uncertainty in the extrapolation of the training data [49, 50].

## Handling incomplete data

Sometimes a database may contain entries with incomplete input information due to experimental design or data acquisition problems. The possibility of such missing data is higher in meta-analysis when results from studies with acceptable differences in design and purpose are pooled to form a database. In our database, for example, the osteochondral defect studies took the area of defect as a common critical data for evaluating the severity of injury [14, 17]. However, this information was not always presented in osteoarthritis studies due to difficulties in precisely measuring defect area with complicated geometry [25, 48]. This leads to "missing data" in the entries. We noticed that underlying correlations may exist across the different properties, and can be distilled by a neural network to "fill in" the missing information. A typical neural network formalism requires each property to be either an input or output of the network, and all inputs must be provided to compute a valid output. In contrast, our neural network takes the known treatment conditions and the therapeutic outcome (if known) as inputs then outputs the predictions for unknown treatment conditions and the therapeutic outcome. Then following the flowchart in Fig 2 the neural network is applied iteratively to cycle the predictions of the unknown treatment conditions and therapeutic outcome until self-consistency, an expectation-maximization algorithm [51].

The algorithm is shown in Fig 2. For any unknown properties, we first set missing values to the average of the values present in the data set. With estimates for all values of the neural network we then recursively apply the following equation until convergence:

$$\mathbf{x}^{n+1} = \gamma \mathbf{x}^n + (1 - \gamma)\mathbf{f}(\mathbf{x}^n), \tag{3}$$

where $n$ denotes the iteration step, $\mathbf{f}(\mathbf{x}^n)$ is a prediction for $\mathbf{x}$ obtained from the neural network. The converged result is then returned instead of $\mathbf{f}(\mathbf{x}^n)$. The function $\mathbf{f}$ remains fixed on each

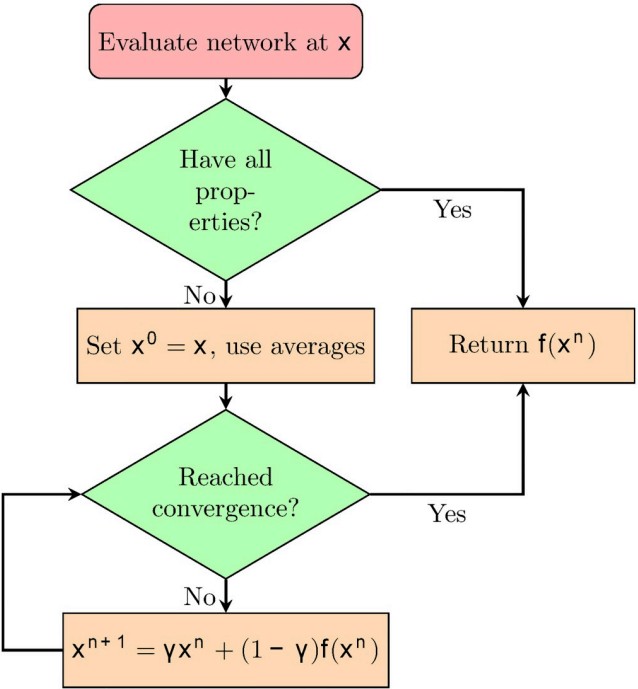

**Fig 2. Data imputation algorithm for the vector x.** After checking the missing properties of entries, we set $x^0 = x$, and replace all the missing data by averages from the training data set. We then iteratively compute $x^{n+1}$ as a function of $x^n$ and $f(x^n)$ until we reach convergence after $n$ iterations.

iteration of the cycle. A softening parameter, $\gamma \in [0, 1]$, is used to combine the results with the existing predictions, and a $\gamma > 0$ serves to prevent oscillations and divergences of the predictions. Typically, we set $\gamma$ as 0.5. The performance against the missing data percentage in the database is shown in S1 Text (Fig. B).

Thus we were able to utilize the full information in the database, derive a more robust model and enhance the quality of predictions.

## Validation

The model was initially fitted on a training dataset, which is a set of entries from our database. The fitted model then used to predict the outputs in a second validation dataset, to provide an unbiased evaluation of the model. To assess the performance of the model, we adapt the coefficient of determination ($R^2$) metric when training our model on the validation dataset. In this work, we are only using the therapeutic outcome as the variable for $R^2$:

$$R^2 = 1 - \frac{\sum_i (y_i^{\text{pred}} - y_i^{\text{actl}})^2}{\sum_i (y_i^{\text{actl}} - \overline{y^{\text{actl}}})^2}$$

where $y_i^{\text{actl}}$ is the therapeutic outcome from the $i$th case (patient or animal) and $y_i^{\text{pred}}$ is the corresponding prediction. The value of $R^2$ ranges from negative infinity to 1 and is a measure of the fit to the perfect identity line $y_i^{\text{pred}} = y_i^{\text{actl}}$, where $R^2 = 1$ means a perfect fit, $R^2 = 0$ corresponds to making the most naive prediction of all values being the average of the data. To confirm the accuracy of the neural network prediction and avoid overfitting, the $R^2$ was calculated within the leave-one-out cross-validation framework. We first removed one entry from the database at a time for all the entries, trained the model on the remaining entries,

and presented the inputs of the unseen entry to predict its output. Eventually we then gathered all the predicted properties of every entry, and computed the $R^2$ against the actual experimental properties.

## Other machine learning methods

We compare our neural network algorithm with a variety of other machine learning approaches in S1 Text (Fig. A(i)). Random Forest (RF) [52] is a popular method, which builds an ensemble of decision trees to predict individual results. However, decision trees require all their input to be present during training that makes it impossible to build RF models using incomplete entries but to drop them, we use the imputation algorithm to fill the database and record the second-best $R^2$ value of 0.554 compared to the value of 0.637 from our neural network method. We have also tested the K-Nearest Neighbor (KNN) and Multiple Linear Regression (MLR) method [53], where 3 nearest neighbors was chosen as the optimal setting of KNN using Euclidean distance.

Another popular method of analyzing sparse databases is matrix factorization, where the matrix of condition and treatment values is approximately factorized into two lower-rank matrices that are then used to predict therapeutic outcome for the new patient. We used the modern Collective Matrix Factorization (CMF) [54] implementation for comparison, and the hyperparameter alpha for the CMF model was chosen heuristically as 0.99. The $R^2$ value is -0.003, the reason might be the CMF method assumes linearity in the interaction of latent factors which fails to capture some complex non-linear interactions.

We also use the leave-one-out cross-validation to determine other hyperparameters of the neural network in S1 Text (Fig. A).

## Selection of input properties

The procedure for the neural network to select the most appropriate input properties is challenging for our meta-analysis, as discussed before the available properties vary across different studies, and the same or related properties may be reported in different ways. The input properties were categorized into two types, factual and derived. The factual properties were: species, implantation cell number, defect area, defect depth, type of cartilage damage, body weight, and tissue source. The type of tissue source can be further classified into bone marrow (BM), adipose tissue (AD), synovial fluid (SF), Warton's jelly (WJ), synovial tissue (ST), and umbilical cord blood (UCB). The derived properties emerged from our biological intuitions and may not have been used in the previous studies, such as defect area percentage, defect depth percentage, and cell concentration.

We first trained a neural network to take only one input property and predicted the cartilage repair score. This allowed us to probe the performance of individual property in Fig 3. It is possible that two or more properties of MSC therapy were individually not impactful to the cartilage repair, but when used in combination they allow the model to capture important correlation. For example, both implantation cell concentration and defect volume have low $R^2$ values (-0.04 and 0.12 respectively), but the implantation cell number, which is the product of two former properties, gives a $R^2$ value of 0.41.

The full set of factual and derived properties was provided as inputs to train the neural network. A correlation test is performed between all properties to make sure no pairs of input properties are closely correlated, finding that both Pearson's Correlation and Spearman's Rank Correlation coefficients are smaller than 0.53. The individual properties' correlation with the cartilage repair score was computed and sorted in descending order in Fig 3. The most correlated property is defect depth percentage, with an $R^2$ value of 0.55, followed by defect area

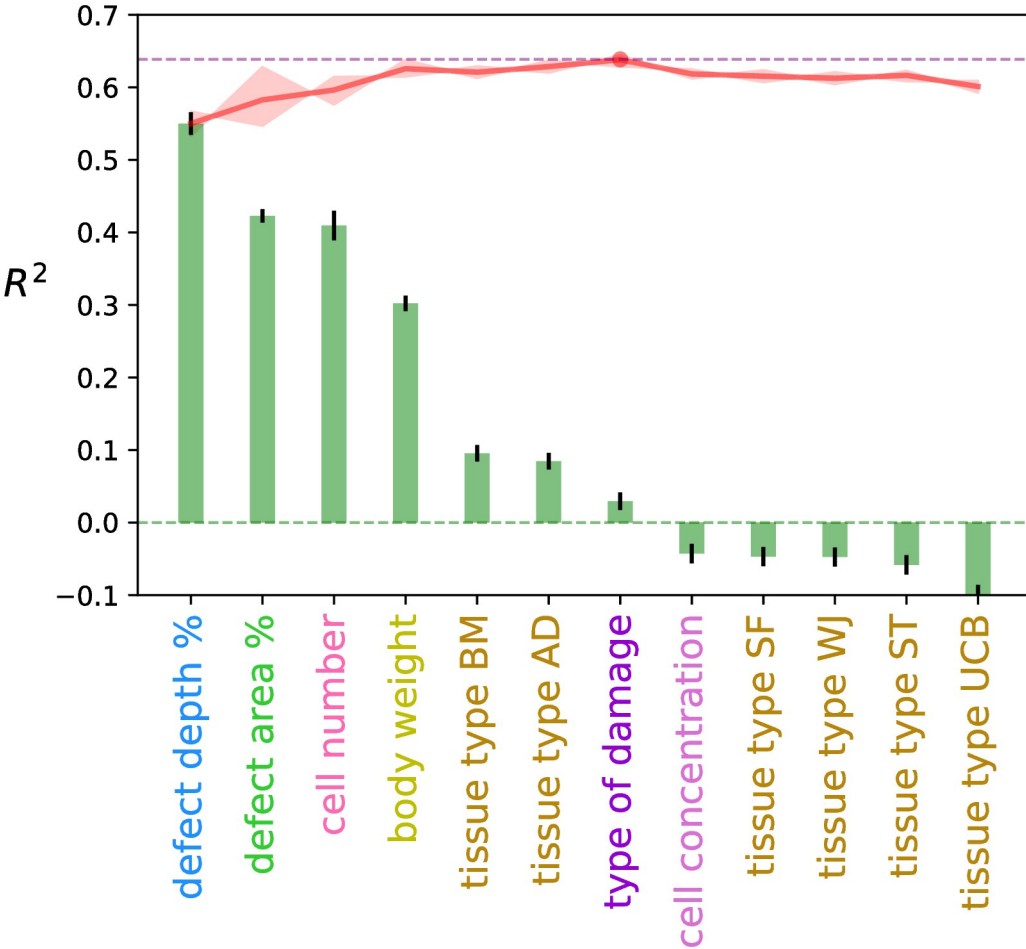

**Fig 3. Accuracy of the neural network.** $R^2$ values of the neural network model trained with individual property (bar) and the combination of best performing individual properties (red line). The shaded red area represents the uncertainty of each $R^2$ value.

percentage (0.42), cell number (0.41) and body weight (0.30). The tissue type BM, AD, and type of cartilage damage are less correlated, and the cell concentration along with other tissue types (SF, WJ, ST, and UCB) are negatively correlated with the cartilage repair score. The top four properties gives an $R^2$ value of 0.625 ± 0.012, and the combination of all seven positive properties has a maximum $R^2$ of 0.637 ± 0.005. Overfitting was observed at a decreasing $R^2$ with more than seven descriptors, this happened when the system matched the training dataset but failed with unseen data draw from the validation dataset. Fewer descriptors did not provide a sufficient basis set, so we chose the first seven descriptors where each of them individually yields a positive $R^2$ value, which captured more correlations of clinical properties without overfitting and provided higher quality uncertainty prediction. The tissue type BM and AD have been consolidated into a single tissue type property, and we have a total of six different input properties.

## Results

With the identified six critical input properties, the neural network used for our machine learning model achieved a $R^2$ of 0.637 ± 0.005 with blind cross-validation. The neural network

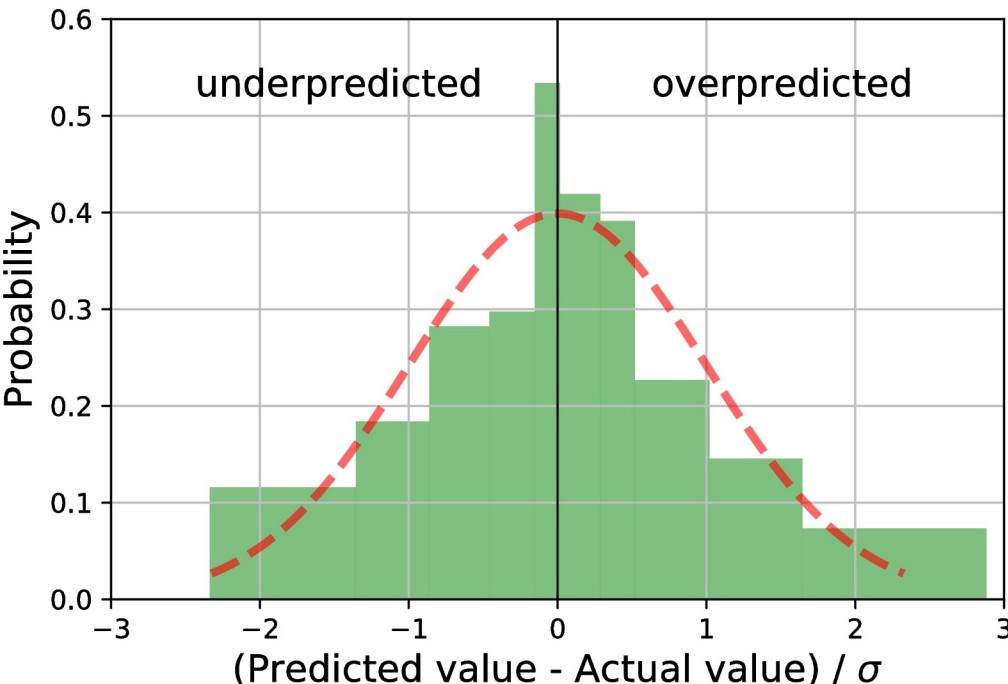

**Fig 4. Histogram of errors for predictions using our neural network.** The dotted red line is fitted with a normal distribution. Each bin contains four data points. Overpredicted refers to predicted values are better than post-treatment cartilage repair scores. Underpredicted refers to predicted values are worse than post-treatment cartilage repair scores.

also delivered the prediction uncertainty in terms of the absolute error between the predicted value and the actual value, as plotted in Fig 4. The random errors associated with the model correctly followed a normal distribution so should well capture the true uncertainty. There are 18 entries out of the total 44 entries lie outside the one standard deviation region. We will exploit all of this knowledge in the next subsection.

## Imputation

With access to the uncertainties in Fig 4, we can gain further insight from the neural network predictions. In particular, we can discard predictions carrying large uncertainty, and trust only those with smaller uncertainty. The idea is illustrated in Fig 5A, where we select four of the points from Fig 5B, including that with the largest uncertainty that has the highest likelihood of deviating from the true value so should give the largest error, as well as other quartiles in uncertainty. This allowed us to focus on the most confident predictions only at the expense of reporting fewer predictions, e.g. discard the data point with the largest uncertainty (yellow bar) and recalculate the sum of squares for the $R^2$ value. By doing so, the quality of the remaining neural network predictions increases as the root-mean-square error between the predicted values and the actual values decreases when a smaller fraction of predictions is accepted and validated as shown in Fig 5B, 100% of data validated means we predicted and validated against every entry in our database, and all of these values contribute to the final $R^2$. 75% of data validated means that we calculate the $R^2$ using only the 75% of the data with smallest uncertainties in their predictions. The best $R^2$ value of 0.743 was reported at 82% of data being validated, and then reached the plateau when less than 70% of data are being validated. Validating fewer data can lead to significant noise and is less applicable in the real-world where we wish to impute as much as possible, therefore we focus on the >50% regime. The result confirms that

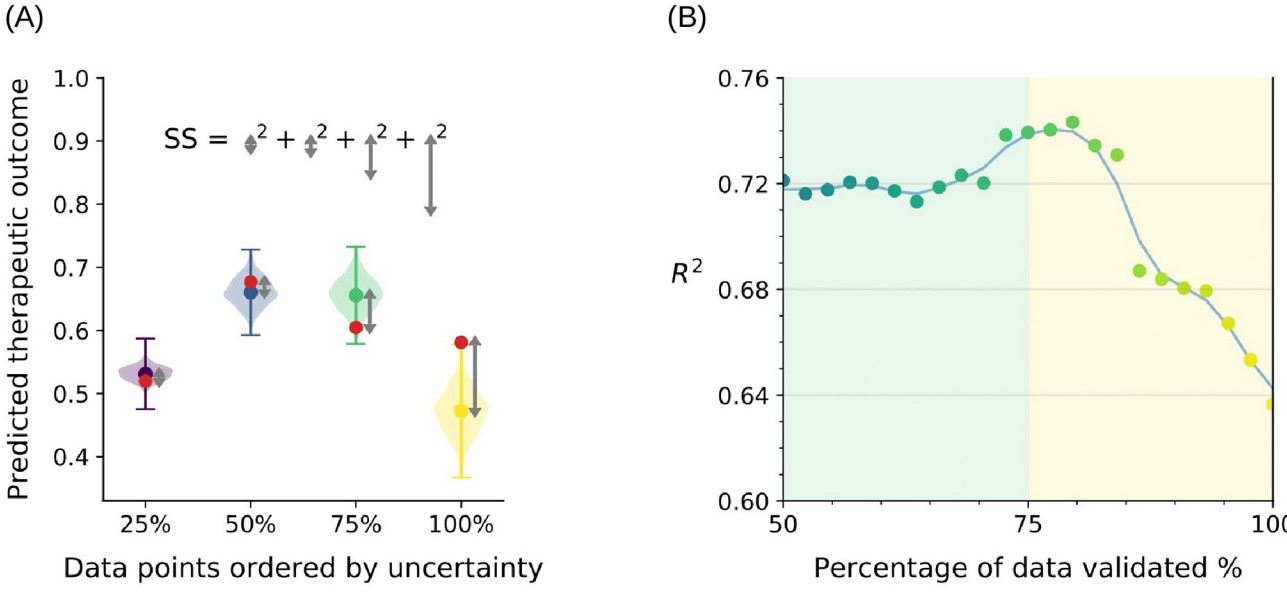

**Fig 5. Model performance after imputation.** (A) shows an example when making predictions for just four data points. The y-axis is the prediction from the machine learning, and the x-axis delineates four different sample predictions ordered by their uncertainty. The colored dots represent the predicted value and their uncertainty that is also predicted by the machine learning method is shown by the colored bars (magenta, turquoise, green, and yellow), the violin plot represents the probability density distribution for predicted outcomes, and the red dots are the true (unknown to machine learning) values used for validation, and the difference between predicted and true values is measured as the grey arrows. The sum of squares (SS) value is then normalized to calculate the $R^2$ value. (B) shows the $R^2$ value with percentage of data validated, and the data points are color-coded by their uncertainty ranking. The blue line is the trend line fitted to the data points. The turquoise, green, and yellow points in (A) are the points at 50%, 75%, and 100% in (B).

the neural network is able to accurately and truthfully inform us about the uncertainties in its predictions; and so the confidence of predictions is correlated with their accuracy.

We note that this post-processing corresponds to increase in accuracy, once a model was trained, and the desired level of confidence can be specified and used to return only sufficiently accurate predictions. The projected cartilage repair score along with the confidence level of the prediction will be provided once the patient's condition has been set as inputs to the model, which will allow clinicians to focus treatments on those most likely to lead to success, and trials to focus on the most illuminating input property space.

## Identifying anomalous results

With the computed uncertainties of prediction, we identified entries with particularly high deviations between the predicted and experimental results. Those can then be re-examined, and corrected to improve the training dataset. Most predictions of our model were expected to lie within one standard deviation (±1) of the experimental results, as shown in Fig 4. The 18 entries lay outside of the one standard deviation region are shown in Table 1. Three of them were from clinical trials, and the other 15 were from animal studies. A positive number of standard deviation away means our neural network overpredicts the cartilage repair score, and a negative number means underprediction. We analyzed the over- and underpredicted repair scores as follows.

In general, our model predicted 80% of the clinical trials with an error smaller than one standard deviation, which was better than that of 48% for the animal studies. Three human clinical trial outcomes were underpredicted. In two of the cases, the researchers performed additional surgical procedures besides MSC implantation to repair the damaged cartilage. De

**Table 1. The table highlights predicted entries where the number of standard deviations out by clinical results are greater than 1 or less than -1, which indicates our prediction is away from the experiments.**

| | Species | Authors | Damage | Standard deviations out by |
|---|---|---|---|---|
| Overpredicted | Rabbit | Katayama et al. [33] | Defect | 2.88 |
| | Rat | Dahlin et al. [38] | Defect | 2.16 |
| | Rat | Papadopoulou et al. [47] | Arthritis | 2.15 |
| | Minipig | Ha et al. [15] | Defect | 1.84 |
| | Rat | Zhu et al. [35] | Defect | 1.35 |
| | Rat | Zhang et al. [43] | Arthritis | 1.20 |
| | Minipig | Wu et al. [17] | Defect | 1.10 |
| | Minipig | Lee et al. [23] | Defect | 1.04 |
| | Rabbit | Li et al. [34] | Defect | 1.02 |
| Underpredicted | Rat | Xue et al. [37] | Defect | -1.03 |
| | Rabbit | Park et al. [14] | Defect | -1.15 |
| | Human | de Windt et al. [18] | Defect | -1.26 |
| | Rabbit | Li et al. [34] | Defect | -1.30 |
| | Human | Koh et al. [22] | Defect | -1.40 |
| | Rabbit | Ma et al. [32] | Defect | -1.62 |
| | Rat | Park et al. [13] | Defect | -1.74 |
| | Human | Fodor et al. [26] | Arthritis | -1.76 |
| | Piglet | Ando et al. [29] | Defect | -2.34 |

Windt et al. implanted debrided autologous chondrocytes together with MSCs in their procedure [18]. The interaction between MSCs and chondrocytes was not considered as an input property in the current neural network, but might promote the cartilage repair. Koh et al. performed microfracture surgery before MSC implantation [22]. The recruitment of autologous MSCs from the subchondral bone to the defect cartilage area by the microfracture surgery was likely the cause of the underpredicted outcome from the neural network.

The most underpredicted entry with -2.34 standard deviations away from the actual experiment outcome, appeared in the study from Ando et al., where an MSC-based tissue scaffold was implanted to chondral defects in porcine models [29]. Similarly, Li et al. encapsulated MSCs in microspheres prior to transplantation to the rabbit osteochondral defects [34], which yielded a standard deviation of -1.30. In both cases, the use of scaffold likely induced pre-differentiation of MSCs towards chondrogenic lineage, and the production of extracellular matrix proteins before transplantation might have greatly promoted the repair efficacy. Xue et al. also delivered MSCs to their rat model in tissue-engineered scaffold made from poly (lactide-co-glycolide) (PLGA)/nano-hydroxyapatite (NHA), but the MSCs possibly remained at undifferentiated status [37]. This resulted in a smaller underprediction by the neural network with a standard deviation of -1.03. Another underprediction with a standard deviation of -1.62 was seen in the study from Ma et al. [32], where an autologous graft was transplanted together with the MSCs. In this study, the mosaicplasty might have contributed significantly to the repair, which was not analyzed as an input to the neural network.

For overpredicted repair scores, Katayama et al. reported their MSC treatment efficacy to rabbit cartilage defect [33] at a much lower level than the neural network prediction, with 2.88 standard deviations away. Although the isolation and subculture of MSCs were performed using standard protocols, the authors did not provide sufficient quality control of the cells before the treatment. The uncertainty in cell purity and quality might have resulted in the suboptimal repair.

Re-visiting these inaccurately predicted cases has allowed us to gain further insights on the therapeutic efficacy of MSCs in cartilage repair. The majority of the less accurate predictions occurred in animal trials, where special delivery methods or manipulations to the MSCs have been implemented. These findings implied the potential impact of these novel therapy input properties on cartilage repair, although they are not readily applied in clinical trials. We also realized that not all the less accurately predicted cases were associated with special delivery strategy or cell modification, and the underlying causes were not obvious. It is reasonable to believe that the potency of MSCs, secretome profile, and the surgical procedures might all impose significant impacts on the therapeutic outcome. Including these information as input properties in the database would empower the neural network to enhance the prediction accuracy.

## Influence of properties

The patients' pre-treatment conditions and therapeutic strategies were encoded within the input properties for the model to make predictions. The relative strength of the properties on predicting the cartilage repair score, defined as the change of $R^2$ on removing a property, is plotted Fig 6. The pre-treatment conditions such as defect area percentage, defect depth percentage, and body weight play important roles in the treatment outcome. Whereas the treatment strategy properties, such as the implantation cell number and the tissue source, impact the outcome to a lesser extent. We now study these input properties in descending order of importance.

**Defect area and depth percentage.** We first investigated the two most important properties: defect area percentage and defect depth percentage; a surface plot is shown in Fig 7 where the cartilage repair score has been normalized against the full range of scores in the database. It is worthwhile to note that although most training dataset has defect area percentage less than 30% and defect depth percentage greater than 40%, our neural network model extrapolated cartilage repair of a patient with indications beyond the existing range of conditions in the database. The neural network can do this due to its unique ability to handle missing data over the full range of conditions (0-100%). In general, the cartilage repair score drops as the percentage of defect area and depth increases, implying the difficulty for MSC therapy to achieve full recovery in patients with severe cartilage damages.

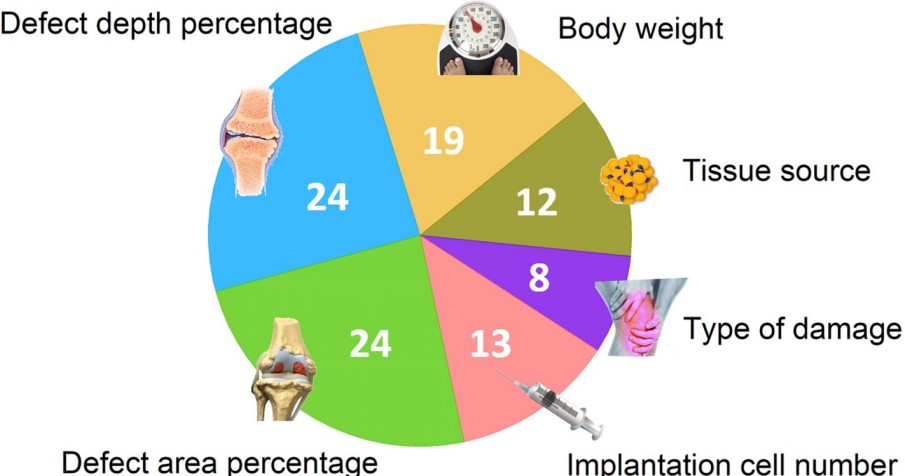

**Fig 6. Illustration of the relative strength of properties used in our model.**

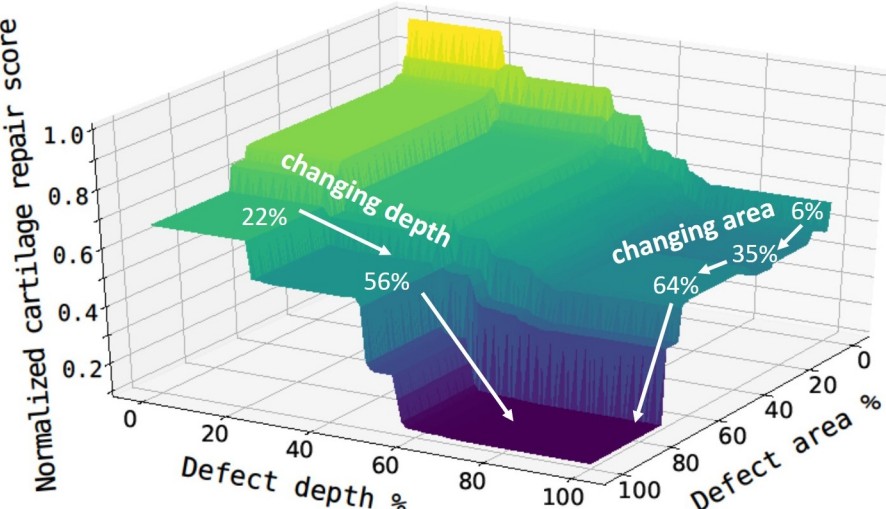

**Fig 7. Surface plot of the normalized cartilage repair score based on defect area percentage and defect depth percentage.** The trajectory of changing area or depth is shown in white arrows.

The study showed that critical thresholds of damage exist for effective cartilage repair to happen, which is similar to the case of volumetric muscle loss [55]. In cartilage repair models, a "critical size" osteochondral defect that can not effectively repair by itself, has been widely used. In most cases, such critical sizes were applied at estimated default values for different animal models. Some studies have attempted to experimentally determine the critical size of the defect in terms of depth and diameter in specific animal models [56]. In our machine learning model, we predicted those "critical size" defects as we observed a rapid decrease in the normalized cartilage repair score when the defect area percentage increases from 6% to 35%. Another fast drop was observed at 64%, because minimal repair should be expected when more than 70% of cartilage area is damaged. These sharp drop-offs identified from the model indicates the presence of multiple "critical sizes" that constrain cartilage repair to different levels post MSC therapy. These quantitative cartilage repair predictions based on the patients' defect conditions provide useful references for the clinicians to make decisions on the therapy.

**Body weight.** As shown in Fig 6, the body weight also acts as an important input property in our neural network: heavier species tend to have a better therapeutic outcome. However, this may be attributed to the large inter-species weight differences in the database. The lack of intra-species weights information in the databased has made further analysis difficult. This could be a valuable topic for further investigation.

**Implantation cell number.** The next most important input property is the implantation cell number. Fig 8 shows a near linear increase in the cartilage repair score with implantation cell number less than 17 million. The normalized cartilage repair score is above 0.9 between 17 to 25 million implantation cell number, and is maintained around 0.8 in the 25 to 75 million range. Further increase in the implantation cell number results in a sudden drop of the normalized cartilage repair score to below 0.7.

The determination of MSC dose for therapy remains intuitive in current practice. A wide range of implantation cell numbers has been found in the literature, ranging from a few thousand to 10 billion with the majority falling between 1 to 100 million [5]. Besides the implantation cell number, these cells were also transplanted at a vast range of concentrations in

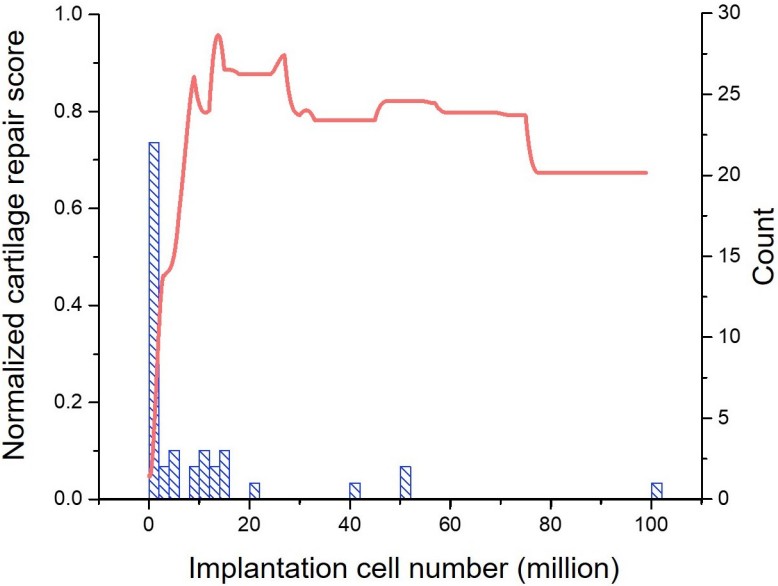

**Fig 8. Impact of implantation cell number on the cartilage repair.** The left y-axis shows the predicted patient's cartilage repair scores normalized to the range of score that patients have been evaluated in clinical trials. The right y-axis shows the number of studies (blue histogram) that use a certain cell number in our database.

different animal studies and clinical trials, between a thousand to a billion cells per millilitre of the delivery agents [5]. Controversial results on the cell dose-dependent influence on cartilage repair have been reported. On one hand, higher cell number and concentration have been associated with better chondrogenesis and cartilage repair [57–62]. The high cell density likely recapitulated the mesenchymal condensation process that occurred during embryonic development of cartilage, and promoted MSC differentiation towards chondrogenic lineage [63]. On the other hand, native cartilage is an ECM-rich avascular tissue with low cell density. Studies have pointed out the limitation to cell saturation and survival [64], and high dose of MSC transplantation was likely to increase the risk of synovitis and synovial proliferation [57, 65].

In this study, we untangle the long-lasting controversy through machine learning approach, and recommend an optimal dose of 17-25 million MSC for human therapy. This conclusion is partly supported by a dose-dependent MSC Phase II clinical trial [48] to treat osteoarthritis patients, which is unseen to the machine learning model, where MSC dose larger than 25 million resulted in a decline in the patients' cartilage repair scores. This overturns the long-standing protocol of using fewer than 2 million cells for implantation.

**Tissue source.** The tissue sources of MSCs, bone marrow (BM) and adipose tissue (AD), have been combined to form a new property in our model. These two sources of MSCs are the most widely used and studied, mainly because of the high accessibility to BM and AD. The abundant MSC number obtainable from BM and AD also determines that these cells have greater potential to be produced at large scale for allogenic uses. The number of occurences of BM and AD MSCs were abundant in our database, and the machine learning results suggested that both BM and AD MSCs are beneficial to the treatment. However, more studies are needed to reach a conclusion on the effects of other tissue sources, including synovia fluid (SF), Wharton's jelly (WJ), synovia tissue (ST), and umbilical cord blood (UCB). Their individual performances were tentatively analyzed and displayed in Fig 3 based on the current database.

**Type of cartilage damage.** The least important property in this machine learning model is the type of cartilage damage. Although fundamental difference exists in the causes and

pathologies between osteochondral defect and osteoarthritis, the mechanisms of cartilage repair through MSC therapy in both cases may share many commonalities, such as differentiation of the MSCs into chondrocytes at the damage site, secretion of regenerative factors, and immune regulation.

## Discussion

In this study, we have developed a neural network model that exploits the inter-property and property-property correlations to predict the therapeutic efficacy of MSC transplantation for cartilage repair based on animal results and human clinical trials. We started with cartilage injury models where different MSCs were given and measures of their performance were recorded. We characterized the cartilage repair score and filled the missing information using the neural network while training the model. The assessment of new patient would provide input information for the model to make predictions on human clinical trial outcomes and the recommended properties, clinicians would be given the uncertainty in the prediction along with the confidence level to decide the most suitable therapy for treatment.

We reported an optimal implantation cell number of 17-25 million to treat patients with cartilage damages, and quantitatively demonstrated how the key factors, including the number of cells implanted, defect area, and depth, could impact the post-transplantation healing. In particular, the neural network has the ability to systematically estimate the confidence level of each prediction, make decisions based on reliable results, and expedite trials. The predictive power of our model enables personalized therapy. We predicted the optimal therapeutic outcome based on individual patient's disease conditions, including defect area percentage, defect depth percentage, and body weight. For patients with severe cartilage damages beyond the threshold for effective repair, other treatment strategies should be considered. Together, the predictions from our model would serve as important references to the clinicians and scientists to design better MSC therapy strategies for cartilage repair, and their findings can be used to further optimize the model. The technology can also be adapted for MSC therapies to other medical indications, and address other biomedical questions.

There is open access to the data and codes at https://doi.org/10.17863/CAM.52036.

## Supporting information

**S1 Text. We provide additional details, including the algorithm to calculate uncertainties and figures that validate the hyperparameters for our machine learning method.** (PDF)

## Author Contributions

**Conceptualization:** Yu Yang Fredrik Liu, Yin Lu, Steve Oh, Gareth J. Conduit.

**Data curation:** Yu Yang Fredrik Liu, Yin Lu, Gareth J. Conduit.

**Formal analysis:** Yu Yang Fredrik Liu, Gareth J. Conduit.

**Funding acquisition:** Steve Oh, Gareth J. Conduit.

**Investigation:** Yin Lu, Steve Oh.

**Methodology:** Yu Yang Fredrik Liu, Yin Lu, Steve Oh, Gareth J. Conduit.

**Project administration:** Yu Yang Fredrik Liu, Steve Oh, Gareth J. Conduit.

**Resources:** Steve Oh, Gareth J. Conduit.

**Software:** Yu Yang Fredrik Liu, Gareth J. Conduit.

**Supervision:** Steve Oh, Gareth J. Conduit.

**Validation:** Yu Yang Fredrik Liu, Yin Lu, Gareth J. Conduit.

**Visualization:** Yu Yang Fredrik Liu.

**Writing – original draft:** Yu Yang Fredrik Liu, Yin Lu.

**Writing – review & editing:** Yu Yang Fredrik Liu, Yin Lu, Steve Oh, Gareth J. Conduit.

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
