## [Decision Letter · Decision Letter 0]

13 Mar 2020

Dear Mr Liu,

Thank you very much for submitting your manuscript "Machine learning mesenchymal stem cell efficacy for cartilage repair" for consideration at PLOS Computational Biology.

As with all papers reviewed by the journal, your manuscript was reviewed by members of the editorial board and by several independent reviewers. In light of the reviews (below this email), we would like to invite the resubmission of a significantly-revised version that takes into account the reviewers' comments.

We cannot make any decision about publication until we have seen the revised manuscript and your response to the reviewers' comments. Your revised manuscript is also likely to be sent to reviewers for further evaluation.

Sincerely,

Qing Nie

Associate Editor

PLOS Computational Biology

Arne Elofsson

Deputy Editor

PLOS Computational Biology

Reviewer's Responses to Questions

**Comments to the Authors:**

Reviewer #1: In this manuscript (m.s.), the authors proposed a neural network model to predict the efficacy of MSC therapies in cartilage repair. The training data was collected from 36 published articles on PubMed. Compared with standard neural network approach, the authors claimed the novelties of their methods lied in the ability to 1) impute missing data and 2) quantify the uncertainty in prediction. The model identified several significant factors that affect cartilage repair, and suggested the optimal selection of MSC in treatment. The m.s. also speculated that the method can be applied to other clinical studies.

Overall, the m.s. addresses the important biological problem of cartilage repair prediction and contributes an associated machine learning method with certain new features. However, to improve the soundness of their conclusions and enhance readability for the PLoS CB audience, a few technical issues should be stressed as follows prior to the recommendation of publication.

Major Points

1. On the presentation of methods.

It leaves me the impression that some necessary technical details are omitted in the current m.s., which should be included in the Methods part, or provided as Supplementary Information.

1) The authors claimed uncertainty-quantification and missing data imputation are the unique features of their proposed method. Unfortunately, I am not able to check such claim after reading the Methods part of the m.s., since they are currently presented in a highly abstract and rough way, and the details are not provided even in the Supplementary. For example, in line 103-109, to describe the uncertainty-quantification procedure, they wrote,

“Several separate networks were trained on the data with different weights, and their variance was a measure of uncertainty in the predictions which accounts for both experimental uncertainty in the underlying data and the uncertainty in the extrapolation of the training data [49,50]. This concept is similar when estimating the uncertainty in ensemble models, with the underlying model being changed to neural networks and the uncertainty estimates generated accurately represent the observed errors in the prediction.”

How can different network be obtained by solving a simple minimization problem? How are the weights determined? How many networks are actually trained in author’s data and what about the robustness? Can certain equations/mathematical notations be used to replace such narrative sentences?

in line 103-109, when describing the imputation method, they wrote

“We treated all properties as both inputs and outputs of the model and adapt an expectation-maximization algorithm to exploit the relationship across the properties using an iterative approach. ”

What is the objective function of EM algorithm in current setting? Does the iteration finally converge in real-data implementation? Can the iteration procedure be explicitly written in pseudo-code or mathematical expressions?

Therefore, I recommend the authors to revise the writing of methods part and associated SI by adding more details about implementation procedures, making it possible for readers to directly inspect the rationale/plausibility of their methods from the methodology perspective.

2) Implementation details of neural network & reproducibility issue

The m.s. reported the R-square regarding the neural network performance, while seems to neglect reporting other key parameters. I suggest the authors to disclose other parameters and perhaps discuss robustness in Supplementary material (such as learning rate, robustness for hidden layer number and layer width, robustness for activation function (ReLU v.s.tanh) and trained weights), which is generally vital to performance of machine learning techniques. I also think it would be very helpful to make the training codes publicly available for reproducible check and further benchmarking/ application, which is quite common for machine learning algorithms.

2. Validation of the imputation function

As mentioned above, the imputation function was alleged to be the highlight of current m.s. For validation of the effectiveness, I recommend the authors to conduct further computational experiment regarding imputation values. For instance, the authors may treat some known values as missing data in the cartilage data, and then evaluate the performance. I notice similar validation has been done in material data by some of the authors in reference [10], which can also be done here to strengthen the claim of current m.s..

3. Comparison with other state-of-art machine learning methods.

The sample size of cartilage repair data used in this m.s. seems not to be overwhelmingly large, therefore neural network method might not out-perform other classical machine learning approaches by default. Since the m.s. used the word ‘Machine Learning’ instead of ‘Neural Network’ in the article title, I suggest the authors to evaluate the performance of other powerful machine learning methods for continuous variables (e.g. random regression forests) to predict the outcome and compare the results (for instance, R-square in testing dataset) with neural network approach, hence highlighting neural network as the desired machine learning method and providing a baseline R-square for comparison.

Minor Points

1. On Line 40, it stated that ‘We adapt a deep learning method ’, which seems inaccurate. In fact, the authors only used very simple single-layer neural network in the current m.s., far from the widely perceived ‘deep learning’ that has multi-layer structure and huge amount of parameters.

2. On line 91, the expression might be f(x)=y instead of f(x)=x.

Reviewer #2: Liu et al. present a neural network model to predict cartilage healing via MSC therapy. The model is trained on public data and its analysis and results are of interest both from methodological and biological perspective. There are a number of serious concerns I have regarding the detail given in the paper and the presentation of results that are important to be addressed. These are listed below.

Major comments

1. Methods: not enough detail, need to provide more details (some could be in supplement), e.g. why was this network topology chosen, number of hidden nodes. “adapt an EM algorithm” - details of this?

2. Methods validation: It seems that this network topology successfully handles missing data by the leave-one-out training strategy. Some discussion of this strategy is needed. I.e. since the input data set is relatively small, would other imputation methods by (e.g.) matrix factorization, followed by a single hidden layer rather than several separate networks, work equally well or better? How does performance of this model compare to alternatives that may not handle missing data explicitly but could do so implicitly (e.g. a variational autoencoder)?  

3. Lack of reproducibility: in addition to the lack of written details on methodology, no source code nor any of the data used to train the model is provided. This is recognized as a growing concern in ML for biomedical studies (see https://arxiv.org/abs/2003.00898). For this study to be helpful to others, and to follow PLOS guidelines, working code should be provided along with documentation such that it can be followed by others. Data used as input could also be summarized in a supplement.

4. Seven features are reported as provided best predictions, but the difference between 7 and 4 features seems to be very small. What is the R^2 for 4 features? Along with its variance via cross-validation? Are 4 predictors worse than 7?

5. How closely correlated are the defect depth and defect area? And what possible effects may this have on predictions?

6. More explanation of Fig. 4 is needed. It is not at all clear what is being presented here? RMSE zero without imputation and increases with imputation?

7. In Fig. 6, sharp drop offs are observed for both parameters. Is this reflecting a real biological prediction or is it due to lack of enough data to “fill-in” - I.e. make more continuous - the landscape in fig 6?

Minor comments

- Fig. 5: 3D pie charts are not good ways of representing data as they skew the actual pie chart! (“closer” slices look bigger) Please choose alternative method for visualizing. 2D pie charts are ok.

- Fig. 8 roadmap - this figure does not seem pertinent to the work done in the paper. Does not contain info relevant to the work done. Update or remove.

- Please edit thoroughly and carefully for English language errors, eg plural of “mechanism” incorrect in multiple places,

- Line 267 typo “we now studying”

**Have all data underlying the figures and results presented in the manuscript been provided?**

Reviewer #1: No: The numerical data that underlies graphs or summary statistics has not been provided in spreadsheet form. I also have not found the specific training data link at https://www.openaccess.cam.ac.uk as provided by the author.

Reviewer #2: No: Summary of the data mined from literature and details of how this was done is not provided.

PLOS authors have the option to publish the peer review history of their article (what does this mean?). If published, this will include your full peer review and any attached files.

Reviewer #1: No

Reviewer #2: No
---

## [Decision Letter · Decision Letter 1]

1 Jun 2020

Dear Mr Liu,

Thank you very much for submitting your manuscript "Machine learning mesenchymal stem cell efficacy for cartilage repair" for consideration at PLOS Computational Biology. As with all papers reviewed by the journal, your manuscript was reviewed by members of the editorial board and by several independent reviewers. The reviewers appreciated the attention to an important topic. Based on the reviews, we are likely to accept this manuscript for publication, providing that you modify the manuscript according to the review recommendations.

Sincerely,

Qing Nie

Associate Editor

PLOS Computational Biology

Arne Elofsson

Deputy Editor

PLOS Computational Biology

[LINK]

Reviewer's Responses to Questions

**Comments to the Authors:**

Reviewer #1: In the resubmitted m.s., Liu et al. improved the presentation of method details -- especially on the data imputation procedure, and included benchmarking results with other machine-learning algorithms. Overall, I appreciate the efforts by the authors to address the major concerns raised in my last review. However, prior to the final recommendation of publication, I suggest some confusions/ambiguities to be clarified in the current revised manuscript.

1. Confusions about the input/output of network

In the clean version of m.s. line 69-71, it is said that the input properties are the various factors, and output properties are outcomes measured by scores. However, in line 134, it states all properties are both inputs and outputs of the models. I suggest the authors to be more specific and explicit here, eliminating the inconsistency, since general readers are very likely to have the same confusion with me.

I recommend to clarify the exact meanings of the variables in Figure 1, in the concrete text of cartilage repair data – for example, the vector x simply represents various factors in data, or both factors and outcomes? And the authors may explain in detail how to predict outcomes when using only the factors given a new test data? Also, when computing R-square, what variables are used?

2. More details about variable selections in benchmarking

When comparing with other methods, the m.s. seems not state how the features are selected optimally for linear regression or KNN. From Methods and Figure 3, we know variable selection is important to improve the R-square of authors’ model. For sake of fair benchmarking, I suggest the authors may also tune the parameters or especially do feature selections for other machine learning algorithms. Also, the RF algorithm can run with incomplete data using some naïve imputation strategies, instead of author’s approach, which might also be incorporated into comparison. These details should be clearly recorded in the manuscript or SI to improve the reproducibility and soundness of the paper.

3. Accessibility of data/codes

By the time of submitting the review report, I still cannot get access to the data and codes using provided webpage or placeholder doi. It shows the doi not found, and I cannot find where to enter the ID on https://www.openaccess.cam.ac.uk/. I will be grateful if the authors can provide me more instructions to find the data, or the direct link to the database.

Reviewer #2: In this revision, the authors have greatly clarified several points and the manuscript is much improved as a result. Below are a couple concerns that I think still need to be addressed.

1. Fig 5A is not at all clear, much more detail needed, e.g. “Values” and “Data points” and not meaningful axis labels. Is the point that the data points are ordered by uncertainty? Perhaps mark specifically where each point in 5A comes from in 5B? It is also not clear what the relationship is between the uncertainty in estimates and the error (distance between true value and predicted) and what the significance of this is?

2. “The data and codes are available at www.openaccess.cam.ac.uk . ( ID: 6CB38B01-7F11-4041-A4F6-74F863C31946 and p laceholder DOI link: " ext-link-type="uri" xlink:type="simple">https://doi.org/10.17863/CAM.52036"

Neither the doi link nor the ID in openaccess.cam database work for me. Thus there is still no available code associated with this manuscript.

**Have all data underlying the figures and results presented in the manuscript been provided?**

Reviewer #1: No: By the time of submitting the review report, I still cannot get access to the data and codes using provided webpage or placeholder doi. It shows the doi not found, and I cannot find where to enter the ID on https://www.openaccess.cam.ac.uk/. I will be grateful if the authors can provide me more instructions to find the data, or the direct link to the database.

Reviewer #2: No: authors say that a database is provided, but the link doesn't work

PLOS authors have the option to publish the peer review history of their article (what does this mean?). If published, this will include your full peer review and any attached files.

Reviewer #1: No

Reviewer #2: No
---

## [Decision Letter · Decision Letter 2]

9 Jul 2020

Dear Mr Liu,

Thank you very much for submitting your manuscript "Machine learning to predict mesenchymal stem cell efficacy for cartilage repair" for consideration at PLOS Computational Biology. As with all papers reviewed by the journal, your manuscript was reviewed by members of the editorial board and by several independent reviewers. The reviewers appreciated the attention to an important topic. Based on the reviews, we are likely to accept this manuscript for publication, providing that you modify the manuscript according to the review recommendations.

Sincerely,

Qing Nie

Associate Editor

PLOS Computational Biology

Arne Elofsson

Deputy Editor

PLOS Computational Biology

[LINK]

Reviewer's Responses to Questions

**Comments to the Authors:**

Reviewer #1: In this revision, the authors have addressed most of my concerns regarding the scientific aspects for the manuscript, except that they do not perform the comparison with random forest + naïve imputation strategy, leaving it as a future work. I appreciate their efforts, and will not insist them to perform such specific task, as long as they make the codes publicly available and reproducible for readers to do the benchmarking themselves. However, after reviewing the submitted data and codes (I was not able to access them until this round of revision), I find that the reproducibility issue should be brought into attention before the recommendation of acceptation.

In the files provided by authors, I can only find a simple python file defining the class of Neural Network, which is quite routine and not novel. I cannot figure out how the data imputation (EM algorithm), claimed as the major contribution and novelty in current work, is achieved in the code. Nor do the authors provide script to reproduce their key findings in the manuscript using the cartilage repair database.

Note it is the policy of the PLoS CB that “authors must clearly provide detail, data, and software to ensure readers' ability to reproduce the models, methods, and results”. The reproducibility issue is especially important in the field of machine learning. I therefore strongly recommend the authors to at least provide 1) the script to reproduce their key results in the manuscript using their dataset, 2) brief tutorials or documentations about their defined functions or class in a user-friendly way, making it convenient for the interested readers to directly implement the algorithm (especially data imputation and uncertainty quantification) in other datasets and do their desired benchmarking to validate the algorithms. Overall, I do think that open, reproducible scripts and clear documentations about the proposed algorithms are necessary before this manuscript is accepted.

Reviewer #2: The reviewers have addressed all of my concerns.

**Have all data underlying the figures and results presented in the manuscript been provided?**

Reviewer #1: **No: **datasets--yes, numerical data that underlies graphs in spreadsheet-- no, reproducible codes -- not convinced

Reviewer #2: Yes

PLOS authors have the option to publish the peer review history of their article (what does this mean?). If published, this will include your full peer review and any attached files.

Reviewer #1: No

Reviewer #2: No
---

## [Editor Report · Decision Letter 3]

20 Aug 2020

Dear Mr Liu,

We are pleased to inform you that your manuscript 'Machine learning to predict mesenchymal stem cell efficacy for cartilage repair' has been provisionally accepted for publication in PLOS Computational Biology.

Best regards,

Qing Nie

Associate Editor

PLOS Computational Biology

Arne Elofsson

Deputy Editor

PLOS Computational Biology

---

## [Editor Report · Acceptance letter]

30 Sep 2020

PCOMPBIOL-D-20-00130R3 

Machine learning to predict mesenchymal stem cell efficacy for cartilage repair

Dear Dr Liu,

I am pleased to inform you that your manuscript has been formally accepted for publication in PLOS Computational Biology. Your manuscript is now with our production department and you will be notified of the publication date in due course.

With kind regards,

Laura Mallard
